# Upper Limb Disorders in Catering Workers

**DOI:** 10.3390/diseases11010012

**Published:** 2023-01-17

**Authors:** Concetto Giorgianni, Francesco Principato, Giovanna Spatari

**Affiliations:** Department of Biomedical Sciences, Policlinico Universitario Messina, University of Messina, 98124 Messina, Italy

**Keywords:** handling weight, work-related musculoskeletal disorders, occupational health hazards

## Abstract

Background: The literature reports that catering workers are exposed to various occupational health hazardsé. Objective: This study aims to assess a cohort of catering workers in relation to upper limb disorders, thus contributing to the quantification of work-related musculoskeletal disorders in this occupational sector. Methods: Here, 500 employees, of which 130 were males and 370 were females, with an overall mean age of 50.7 years and an average length of service of 24.8 years, were examined. All subjects completed a standardized questionnaire: the medical history questionnaire of diseases of the upper limbs and spine proposed in “Health surveillance of workers”, third edition, EPC. Results: The obtained data enables the following conclusions to be drawn. Musculoskeletal disorders affect a wide range of catering workers. The most affected anatomical region is the shoulder. These disorders increase with advancing age, specifically shoulder, wrist/hand disorders and daytime and nighttime paresthesias. Employment seniority in the catering sector increases the likelihood of all considered conditions. An increase in weekly workload exclusively affects the shoulder region. Conclusions: This study aims to serve as an impetus for further research that seeks to better analyze musculoskeletal problems in the catering sector.

## 1. Introduction

A study of literature [1,2,3,4] and the European Agency for Safety and Health at Work [5] showed that workers pertaining to the HoReCa (hotels, restaurants and catering) sector are subject to multiple risk factors strongly correlated with the onset of musculoskeletal disorders. Following exposure to these risk factors, approximately 33% of European workers complain of back pain, 20.3% of neck pain, 11.5% of pain to the upper limbs and 17.6% of lower limb pain.

The literature, although scarce, reports that those involved in the catering business are exposed to various occupational hazards linked to their specific tasks [3,6,7,8]. Such hazards may promote the onset of symptoms, more or less disabling, that are specifically linked to the musculoskeletal system.

In a systematic review, Xu et al. [9] investigated the relationship between the prevalence of musculoskeletal disorders and risk factors in the catering industry, highlighting the need for more epidemiological data. The most important risk factors were physical job demands, such as work posture, force applied and repeated movement.

In keeping with these considerations, Bonzini et al. [1] studied musculoskeletal disorders in large-scale food distribution and catering services, noting that although they represent the leading cause of disease and absence from work, there is, nevertheless, a relative scarcity of prospective and case-control studies.

The same authors also stressed a possible selection bias, as some studies only considered data collected in case studies conducted on selected subjects (i.e., those seeking medical attention with regard to a problem) rather than the entire population of workers in the industry. This selection bias may render the evaluation of occupational health risks inaccurate and perhaps distorted. To avoid such potentially skewed assessments, the presence of an occupational physician and active health surveillance aimed at detecting work-related illnesses is warranted.

In 2019, in a cross-sectional study of 300 catering workers, Jahangiri [10] showed that as many as 70% complained of musculoskeletal disorders, with particular attention to the upper limbs.

Tegenu [11], in 2021, in another cross-sectional study of 633 catering workers, showed that 81.5% of workers had musculoskeletal disorders and that the upper back, elbow and wrist were the most affected parts.

Kerckheve [12], in 2022, in a prevalence study on workers employed in different activities, identified catering workers as showing the highest incidence of chronic pain and stated that this data was justified by a high incidence of repetitive gestures, especially at the expense of the upper limbs.

To our knowledge, the literature is void of studies that investigate gender differences among work-related musculoskeletal disorders in restaurant and catering employees.

This study aims to assess a cohort of catering workers in relation to upper limb disorders, thus contributing to the quantification of work-related musculoskeletal disorders in this occupational sector.

## 2. Materials and Methods

Six hundred and eighty-seven adult subjects (age > 18 years) working in catering or restaurant businesses in various regions of northern Italy were examined. The workers were tested during the health surveillance activity, which is an obligation of the employer introduced by Italian legislation.

All subjects completed a standardized questionnaire, specifically, the medical history questionnaire of diseases of the upper limbs and spine proposed in “Health surveillance of workers” third edition, EPC [13].

In particular, the first part of the questionnaire investigated the presence of disorders and their frequency both weekly and monthly in the last 12 months affecting the shoulder, elbow, wrist and hand; the appearance of both day and night paresthesias and full-blown diagnoses; the therapy carried out and finally the days of absence from work caused by these disorders in order to improve the health conditions of the upper limb.

At this stage of our research, it was decided to investigate exclusively the disorders affecting the upper limb, with the promise of further investigations on what has been highlighted affecting the spine. Furthermore, in a subsequent phase of our research, we wanted to investigate whether the presence of these disorders changed with respect to sex, age, length of service and the characteristics of work rhythms.

Employees who had ongoing disputes with their company and those who had previously been employed for at least five years in sectors classified as being high risk for musculoskeletal disorders (such as construction, health care professions, agriculture, industrial jobs, etc.) were excluded from the study.

The remaining sample consisted of 500 employees, of which 130 were males and 370 females, with a mean age of 50.7 years and an average length of service of 24.8 years.

Each subject, moreover, besides the administration of the questionnaire, underwent a medical examination with clinical and functional musculoskeletal assessment. In particular, the presence of conditions affecting the shoulders, elbows, hands and the occurrence of both diurnal and nocturnal paresthesias were investigated.

The results are presented as averages and standard deviations for the qualitative parameters, while the categorical are presented as absolute frequencies and percentages.

All associations were tested with the Chi-square test.

All statistical analyses were calculated using R software (rel 4.2.0), and all values of *p* < 0.005 were considered significant.

## 3. Results

Below, Table 1 summarizes the main characteristics of the subjects comprising the cohort.

A 73% prevalence of musculoskeletal disorders of the upper limbs was observed among the 500 employees. Of the subjects who reported these disorders in at least one of the anatomical regions of interest (365 subjects), 23% were male subjects, and the remaining 77% were females. Of these individuals, several specified the presence of multiple upper limb disorders in their medical history.

Table 2 illustrates the percentages of subjects who reported disorders in relation to the various anatomical regions of the upper limbs. The analysis of the table shows that the shoulder is the anatomical district most affected by the presence of disorders, followed by the elbow and finally by the wrist and the hand.

The Chi-square test was applied to test whether the differences highlighted in the above table were statistically significant in relation to gender, age groups, length of service or weekly work shifts.

In order to do this, as far as age is concerned, it was decided to group the sample of workers into 2 sub-groups (the first comprising workers <50, and the second ≥50).

With regard to seniority, it was decided to group the sample into 2 sub-groups, the first comprising seniority between 1 and 20 years, and the second with seniority above that.

The values obtained are documented in Table 3.

As can be observed in the latter, significant statistical associations (*p* < 0.05) between gender and shoulder disorders, hand/wrist disorders and nighttime paresthesia were found. Regarding age, significant correlations were found between specific age groups and shoulder disorders, hand/wrist disorders and nighttime and daytime paresthesia. With regard to the length of service, significant correlations were found with all five categories of upper limb disorders. Lastly, weekly work shifts did not result significantly associated with all parameters examined.

Table 3 illustrates the *p*-values for the correlations between the four aforementioned variables and the five musculoskeletal conditions cited in Table 2, as described in detail hereafter.

With regard to the relationship between shoulder musculoskeletal disorders and gender, a significant *p*-value of 0.005 was obtained, with a higher prevalence in males. The relationship between gender and elbow disorders does not appear to be statistically significant, as the resulting *p*-value was 0.096 and thus sensibly greater than 0.05. The *p*-value reflecting the association between gender and disorders of the wrist or hand was 0.004, resulting in a statistical significance, with a greater prevalence in female subjects. The relationship between gender and nighttime paresthesia appeared significant, as the *p*-value was 0.001, with a higher prevalence in males. On the contrary, the correlation between gender and diurnal paresthesia was found to be insignificant, with a high χ^2^ value of 0.819.

However, in the association between age groups and musculoskeletal disorders, a significant *p*-value of 0.001 was attained for shoulder, wrist and hand disorders, as well as for daytime and nighttime paresthesias. Furthermore, age was found to be directly proportional to the aforementioned variables; in fact, disorder prevalence increased with increasing age. The relationship between age groups and elbow disorders was not statistically significant, with the *p*-value being 0.04.

The relationships between length of service subgroups and shoulder disorders, disorders pertaining to the wrist and hand region and diurnal and nocturnal paresthesia were significant since the *p*-value of 0.001 and similar was observed for each correlation. The association between elbow disorders and length of service subgroups was also found to be statistically significant, with a *p*-value of 0.021. The presence of said conditions was augmented with increasing length of service.

As for weekly work shifts, no significant correlation was found with all musculoskeletal conditions, as can be observed in Table 3.

## 4. Discussion

As highlighted by the extant literature, few studies have investigated the prevalence of musculoskeletal disorders among catering workers. Employees of the Far East that were subjected to exceedingly strenuous working conditions were examined. These studies are cross-sectional analyses that have collectively considered symptoms reported by employees pertaining to the previous twelve months of work [9,14,15].

In 2006, Dempsey and Filiaggi [4] identified musculoskeletal disorders in 42 out of 100 restaurant waiters, with higher prevalence in the lower back area (18%) and shoulder (11%). The authors note that these disorders were found, albeit manual handling of loads was not excessive. These results are well correlated with the data obtained in this study, which also identifies the shoulder as the most affected upper limb region. The elevated presence of musculoskeletal disorders found in this study can be attributed to the fact that our cohort endured much more demanding working conditions compared to those described by Dempsey and Filiaggi [4]; furthermore, both the average age and length of service were higher in our sample.

While our data were correlated with what Jahangiri [10] described, who found a 70% disease rate in a cohort of 300 workers, the increase in age and seniority had a significant association with the prevalence of musculoskeletal disorders.

Bonzini et al. [1] affirm that there is a noteworthy risk of developing musculoskeletal disorders in the catering profession, as emphasized by the majority of related studies. Nevertheless, these authors specify that the latter is subject to a number of methodological limitations, a result of cross-sectional evaluations with potential subject-selection bias. In fact, most of the studied cohorts are comprised of volunteers.

The present study considered the limitations identified by Bonzini and his colleagues and, albeit using a cross-sectional survey, a large sample was selected to better represent the variety of employees working in the catering industry; moreover, strict exclusion factors were applied in order to render the studied population as homogeneous as possible.

Subramaniam and Murugesan [16] found that 67.5% of 114 male kitchen workers in South India presented musculoskeletal disorders. However, their cohort reflected a very young population, with an average age of 26.4 years. Upon considering only the oldest age group of workers from their sample, consisting of subjects 41 years or older and employees having 6–10 years of work experience, rates of musculoskeletal disorders were 93.9% and 87.5%, respectively. Although the age and length of service of the latter groups are comparable to the characteristics of our sample, the observed prevalence of musculoskeletal disorders was higher than those recorded by us (73%).

A high incidence of disorders was also recently identified by Tegenu [11], who, in 2021, using a structured Nordic questionnaire similar to ours, highlighted that 81.5% of the tested working population had the highlighted disorders, especially those affecting the shoulder in the last 12 months.

Another study, conducted on a sample of 905 hotel restaurant workers in Taiwan, found that 785 (84%) reported experiencing work-related musculoskeletal disorders in the previous month [2]. In keeping with our data, the highest prevalence rate of such disorders regarded the shoulder (58%), albeit the percent of general musculoskeletal disorders was, once again, higher in this cohort than in ours.

Another study reporting a higher prevalence rate of work-related musculoskeletal disorders in restaurant workers with respect to our findings is that of Yeung et al. [17]. Indeed, these authors reported that 87% of Chinese restaurant workers in Hong Kong suffered from musculoskeletal disorders.

As previously mentioned, we believe these differences are attributable to differing working conditions, which tend to be more intense in the Far East where, in fact, the cited studies were conducted.

With regards to predisposing factors, the few studies present in the literature indicate age, gender, work seniority and number of work hours as significant risk factors for the development of musculoskeletal symptoms, thereby substantiating our findings.

The most indicative risk factor disclosed by our study appears to be work seniority, for its elicited statistically significant correlations with all of the considered conditions.

## 5. Conclusions

The obtained data enables the following conclusions to be drawn:Musculoskeletal disorders affect a wide range of catering workers;The most affected anatomical region is the shoulder;These disorders increase with advancing age, specifically shoulder, wrist/hand disorders and daytime and nighttime paresthesias;Employment seniority in the catering sector increases the likelihood of all considered conditions;The increased workload does not affect all the parameters examined.

Furthermore, with regard to gender differences, there is a greater distribution of shoulder disorders in men and of hand and wrist disorders in women. On the other hand, with regard to paresthesias, only nocturnal paresthesias seem to be more prevalent in men. The obtained data appear promising, but larger, preferably case-control, studies are warranted to validate our findings.

In light of these findings and the data present in the literature, it is clear that the catering and restaurant businesses represent a complex reality—that we know relatively little in terms of risk assessment and health consequences.

Working conditions appear to be greatly variable in different scenarios, and a satisfactory standardization of processes has yet to be established. Owing to this inherent complexity, a more thorough and careful examination of problems related to these processes is indispensable.

This study aims to serve as an impetus for further research that seeks to better analyze musculoskeletal problems in the catering sector.

It is our hope that future research is designed and conducted with the support of an occupational health physician, for the latter represents the most suitable professional figure to conduct risk evaluations, assess healthcare aspects and select preventive measures to ensure the safety of the workers.

## Figures and Tables

**Table 1 diseases-11-00012-t001:** Characteristics of the sample.

Characteristics of the Cohort	(*n* = 500)
Age (years) Male subjects (*n*, %) Female subjects (*n*, %) Length of service (years)	50.7 ± 8.4 130 (35%) 370 (65%) 24.8 ± 8.2
Prevalence of upper limb musculoskeletal disorders (%)	73%

**Table 2 diseases-11-00012-t002:** Percentage of disorders or symptoms concerning specific upper-extremity regions.

Shoulder Disorders		Elbow Disorders		Wrist/Hand Disorders		Nocturnal Paresthesia		Diurnal Paresthesia	
TOT	%	TOT	%	TOT	%	TOT	%	TOT	%
243	**48.6**	152	**30.4**	110	**22**	136	**27.2**	89	**17.8**

**Table 3 diseases-11-00012-t003:** Values reflecting the relationship between selected musculoskeletal conditions and gender, age, length of service and shifts with absolute frequencies of workers with pathologies and *p*-values.

	Shoulder Disorders	Elbow Disorders	Wrist/hand Disorders	Nocturnal Paresthesia	Diurnal Paresthesia
Gender (M–F)	(77–166) 0.005	(32–120) 0.096	(17–93) 0.004	(53–83) 0.001	(24–65) 0.819
Age (divided into subgroups (<50 age ≥50 age)	(67–176) 0.001	(46–106) 0.04	(24–86) 0.001	(27–109) 0.001	(19–70) 0.001
Length of service divided into subgroups (up to 20 years–up to 42 years)	(46–197) 0.003	(31–121) 0.021	(10–100) 0.001	(11–125) 0.001	(9–80) 0.001
Weekly work shifts divided into subgroups 23/33 h–34/43 h)	(147–96) 0.344	(95–57) 0.976	(67–43) 0.678	(79–57) 0.203	(55–34) 0.836

## Data Availability

The data presented in this study are available on request from the corresponding authors.

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
