# Peer review of "Upper Limb Disorders in Catering Workers"

_diseases, 2023, doi:10.3390/diseases11010012_

Round 1
Reviewer 1 Report
The study reports interesting results obtained from a large sample representing a specific productive sector. The data collected by the authors will give the opportunity to improve the safety and health of operators in the catering sector.
Author Response
Thanks

Reviewer 2 Report
The objectives presented in the study are very open (i.e. "This study aims to assess a cohort of catering workers in relation to upper limb dis- 67 orders, thus contributing to the quantification of work-related musculoskeletal disorders 68 in this occupational sector. "), which has to be reworded to be specific to the study conducted. The words "occupational sector" may be replaced by "catering or restaurant businesses"
line 159: in the following and at many other places, it is not clear whether it is p value or the value of chi-square?
“value of 0.000”
Table 3: I suppose these values are not chi-square values but rather p-values from the Ch-square test. Please check it seriously.
Discussion and conclusion may not be in the same heading. Better to have two separate headings.
line 176. to 187 are points for conclusions, which must be put in the end. These conclusion points could not be written before discussion on the results.
line 184-186: it is written that there were more MSDs of the shoulder in men and of wrist/hand in women. Therefore it is important to see the prevalence of the type of task done by men and women, in case they are taking the different tasks, unknowingly. Although, the authors are performing the analysis based on assumptions that all tasks are of one industry, having similar nature.
line 236: try to avoid such words as “we believe these ” and use the third person in writing papers.
In the review as well as in the results of the present study, age is a prominent factor affecting WMSDs, therefore it is important if authors could develop a relationship based on regression/correlation among these factors.
lines 257 & 258: may be part of the introduction instead of in the end of the discussion.
The discussion seems to be very scattered in the presentation, making a logical flow of the questions raised based on results and their cross-validation.
In general, English is to be corrected as also the formatting of the manuscript.
Author Response
1) We have corrected the errors
2) we revised the table and updated all values
3)we have corrected the value, unfortunately we do not have the data of subjects with multiple pathologies
4)we have rewritten Tab 3 with what was requested
5) Done
Reviewer 3 Report
I hope the below opinions may help the manuscript be better.
1.Please follow the writing regulation requests on the journal “disease.” Besides, some typo errors should be amended, such as line 42, 66, 103, 196, 226 and the like.
2.In line 185-187. The authors said that “both diurnal and nocturnal paresthesias appear to be significantly more prevalent in men than in women.” However, according to the Table 3, gender had no significant correlation with diurnal paresthesias with chi-square only 0.819. Besides, were the chi-squared values such as 0.000, 0.043, 0.004, 0.005, 0.021 in red probabilities or chi-squared values? Chi-square tests with so much lower values can not reach significant level. The authors should present both chi-squared values and p-values.
3. In Table 2, for nocturnal paresthesia the ratio is 27.2%, not 27% for 136 divided by 500. Besides, the authors reported ”Of the subjects who reported said disorders in at least one of 112 the anatomical 113 regions of interest (365 subjects), 23% were male subjects and the remaining 77% were 114 females.” Can the authors also report how many subjects and percentage had at least two upper limb musculoskeletal disorders ? such as 30% had shoulder and elbow disorders, 27% had shoulder and Wrist/hand dis[1]orders. 15% had shoulder, elbow, and Wrist/hand dis[1]orders….
4. Number and percentage of subjects of each subgroup (names of the groups) for each condition should be included in the Table 3. That will be clearer to understand the details. For example, number and percentage of men and women in shoulder disorders, elbow disorders,… diurnal paresthesias. Number and percentage of subjects in each Age’s subgroup (42-49 years, > 49 years ) in shoulder disorders, ebow disorders,… diurnal paresthesias….
5. The important information on “a 73% prevalence of musculoskeletal disorders of the upper limbs for catering workers” can be included in the Abstract section.
Author Response
We have rewritten Tab 3 as requested, corrected line 159 and separated the discussion from the conclusions, and clarified the evaluation with respect to age and finally corrected the formatting of the article and revised the English with a native speaker colleague

Round 2
Reviewer 2 Report
corrections are done satisfactorily.
Regards